# In Vitro Screening of Sugarcane Cultivars (*Saccharum* spp. Hybrids) for Tolerance to Polyethylene Glycol-Induced Water Stress

**César A. Hernández-Pérez [1], Fernando Carlos Gómez-Merino [1], José L. Spinoso-Castillo [1] and Jericó J. Bello-Bello [2,*]**

1    Colegio de Postgraduados Campus Córdoba, Km. 348 Carretera Federal Córdoba-Veracruz, Congregación Manuel León, Amatlán de los Reyes, Veracruz 94946, Mexico; hernandez.ceeesar@gmail.com (C.A.H.-P.); fernandg@colpos.mx (F.C.G.-M.); jlspinoso@gmail.com (J.L.S.-C.)
2    CONACYT-Colegio de Postgraduados Campus Córdoba, Km. 348 de la Carretera Federal Córdoba-Veracruz, Congregación Manuel León, Amatlán de los Reyes, Veracruz 94946, Mexico
*    Correspondence: jericobello@gmail.com

**Abstract:** Water stress caused by drought affects the productivity of the sugarcane crop. A breeding alternative is the selection of drought-tolerant sugarcane cultivars. The objective of this study was the in vitro screening of cultivars tolerant to water stress using polyethylene glycol (PEG) as a stressing agent. Cultivars (cv) Mex 69-290, CP 72-2086, Mex 79-431 and MOTZMex 92-207 were subjected to different concentrations of PEG 6000 (0, 3, 6 and 9% *w/v*) using Murashige and Skoog semi-solid culture medium. At 30 days of culture, different developmental variables and dry matter (DM), total protein (TP), proline (Pr) and glycine-betaine (GB) contents were evaluated. The results showed reduced development in cv CP 72-2086, Mex 79-431 and MOTZMex 92-207 with increasing PEG concentration. The cv Mex 69-290 showed tolerance to osmotic stress of −0.45 MPa using 3% PEG. Overall, TP content decreased with increasing PEG concentration, while DM, Pr and GB contents rose with increasing PEG concentration in all evaluated cultivars. Our results suggest that cv Mex 69-290 has a slight tolerance to water stress and could be used for rainfed cultivation with low rainfall or reduced irrigation for better water use efficiency. In conclusion, the in vitro screening technique of cultivars tolerant to PEG-induced water stress is an alternative for early determination of drought stress in sugarcane.

**Keywords:** water stress; PEG 6000; proteins; proline; glycine-betaine

## 1. Introduction

Sugarcane (*Saccharum* spp. Hybrids) is a globally important crop because it provides about 80% of the sugar consumed worldwide. It is also used for the production of bioethanol and other byproducts [1,2]. For sugarcane, drought caused by climate change directly affects crop productivity [3,4]. An alternative to address this problem is the use of plant biotechnology techniques, such as plant tissue culture (PTC), which is a collection of tools for genetic improvement in plants [5]. In vitro selection using PTC techniques is an alternative for identifying sugarcane cultivars tolerant to water stress. In vitro selection of water stress-tolerant plants using polyethylene glycol (PEG) is a commonly used strategy in plant biotechnology [6]. PEG acts as an osmotic agent by reducing water availability in the culture medium. High molecular weight PEG (6000–8000) is commonly used as a selective agent because it does not penetrate plant tissues, which makes it ideal for the generation of water stress in culture media under in vitro conditions [7]. Water stress causes different response mechanisms in plants, such as alterations in development, synthesis of specific proteins and, osmotic adjustment mediated by ions and compatible osmolytes [8].

The use of PEG has been studied under in vitro conditions in species such as tomato (*Lycopersicum esculentum* L.) [9], cocoa (*Theobroma cacao* L.) [10], maize (*Zea mays* L.) [11],

chickpea (*Cicer arietinum* L.) [12] and mango (*Mangifera indica*) [13]. In the family Poaceae, it has been studied in species such as rice (*Oryza sativa* L.) [14], sorghum (*Sorghum bicolor*) [15] and wheat (*Triticum aestivum* L.) [16]. In sugarcane, in vitro selection for tolerance to salt (NaCl) and PEG 8000 was evaluated in cv CoC671 [17]. Drought tolerance potential was evaluated in elite genotypes and progenies of sugarcane [18]. Gamma ray-induced mutagenesis and in vitro selection of sugarcane plants tolerant to NaCl was achieved for cv Co740 embryogenic calli [19]. Accessions of different sugarcane species (*Saccharum* spp., *S. robustum*, *S. officinarum*) regarding tolerance to in vitro salinity were evaluated [20]. This study aimed to develop an in vitro selection system for cultivars of *Saccharum* spp. Hybrids tolerant to water stress induced by PEG 6000 as a selective agent.

## 2. Materials and Methods

### 2.1. Plant Material Selection and In Vitro Establishment

Sugarcane (*Saccharum* spp. Hybrids) plants of cultivars Mex 69-290, Mex 79-431, CP 72-2086 and MOTZMex 92-207 were collected from the germplasm bank of the Colegio de Postgraduados Campus Córdoba in Veracruz, México. The most important cultivars for sugarcane production in Mexico are CP 72-2086, Mex 69-290 and Mex 79-431, which represent around 70% of the cultivated area, while MOTZMex 92-207 is a promising cultivar with high productive potential. Apices of 30 cm in length were taken, wrapped in paper bags and kept for 1 day under refrigeration at 4 °C, after which they were reduced to a length of 15 cm and placed in thermohydrotherapy using a digital thermostat (Ecoshel, SC-15, Pharr, TX, USA) at 50 °C for 20 min. Subsequently, the apices were immersed for 1 min in 70% (*v/v*) ethanol and moved to a laminar flow hood where they were reduced to 1.5 cm in length and then transferred to 0.54% (*w/v*) NaClO solution for 10 min and rinsed twice with sterile distilled water. Finally, apical meristems were planted individually in $22 \times 150$ mm test tubes containing 10 mL of MS [21] semi-solid medium supplemented with 30 g L$^{-1}$ sucrose without growth regulators and 0.22% (*w/v*) Phytagel™ (Sigma-Aldrich®; St. Louis, MO, USA) as a gelling agent. The pH of the culture medium was adjusted to 5.8 with 0.1 N NaOH and autoclaved for 15 min at 120 °C and 117.7 kPa. The explants were incubated at $24 \pm 2$ °C with a photoperiod of 16 h light with white LED lamps, with an irradiance of $40 \pm 5$ µmol m$^{-2}$ s$^{-1}$. After one week of culture, the meristems were transferred for the multiplication phase to MS medium supplemented with 30 g L$^{-1}$ sucrose, 1 mg L$^{-1}$ kinetin, 0.6 mg L$^{-1}$ indoleacetic acid and 0.6 mg L$^{-1}$ benzylaminopurine. All reagents were purchased from Sigma-Aldrich®; St. Louis, MO, USA.

### 2.2. In Vitro Selection Pressure with PEG

For in vitro selection pressure with PEG, 2 cm long shoots of the selected cultivars obtained after three subcultures (30 d each) in the multiplication stage were used. The shoots were cultured in $22 \times 250$ test tubes containing MS semi-solid medium without growth regulators and different concentrations of PEG 6000 SIGMA™ (0, 3, 6 and 9% *w/v*). The pH of the culture medium and the sterilization and incubation conditions were the same as described above. Each treatment consisted of 10 explants, with one shoot per test tube. At 45 d of culture, shoot length, number of leaves, root length, number of roots and percentage of dry matter were evaluated. Dry matter content was calculated using dry weight/fresh weight $\times$ 100. Dry weight was determined after placing the shoots in a drying oven (Felisa, FE292, JAL, MX) at a temperature of 75 °C for 72 h. In addition, the osmotic potential of the culture medium was measured and the total protein (TP), proline (Pr) and glycine-betaine (GB) contents were determined in all treatments.

### 2.3. Osmotic Potential Measurements of the Culture Medium

The osmotic potential ($\Psi$s) of the culture medium containing different concentrations of PEG 6000 was determined by the dew point principle using a vapor pressure osmometer (Wescor VAPRO, South Logan, UT, USA).

## 2.4. Total Protein Estimation

TP estimation was carried out by the method proposed by [22]. First, 10 mg of dry plant material were weighed and macerated in a mortar in 25 mL of cold acetone for 5 s. The macerated tissue was vacuum filtered until obtaining acetone powder. Then, 1.25 mL of 0.1M tris-HCl buffer pH 7.1 was added to the resulting powder, which was placed on ice. Subsequently, the solution was centrifuged at 3100 RCF for 20 min at a temperature of 4 °C. Finally, 5 mL of Bradford solution was added to a 0.1 mL sample of the supernatant and read at an absorbance of 595 nm in a spectrophotometer (Thermo Scientific Genesys 10S, Chelmsford, MA, USA). Quantification was done using a calibration curve with bovine albumin.

## 2.5. Proline Determination

Pr was estimated according to the colorimetric method described by [23]. First, 50 mg of fresh leaf tissue were acquired, macerated in a mortar and homogenized with 5 mL of 3% sulfosalicylic acid. The resulting paste was filtered (Whatman #2) and a 1 mL aliquot was taken from the liquid obtained, to which 1 mL of glacial acetic acid and 1 mL of ninhydrin were added. This mixture was mixed for 15 s and left to incubate in a thermoregulated bath for 1 h at 100 °C. The tubes were removed and rapidly chilled on ice. Then 2 mL of toluene was added and mixed for 30 s. The upper phase (toluene and colored complex) was removed with a Pasteur pipette. The absorbance of the resulting chromophore was read at 520 nm in the spectrophotometer (Thermo Scientific Genesys 10S, Chelmsford, MA, USA). The values were interpolated in a calibration curve using a L-proline standard.

## 2.6. Glycine-Betaine (GB) Determination

The determination of GB was carried out with the colorimetric method proposed by [24]. First, 50 mg of dry macerated plant tissue were acquired to which 2.5 mL of deionized water was added. The mixture was filtered through Whatman #2 filter paper and an aliquot of 1 mL diluted at a 1:1 ratio with 2 N $H_2SO_4$ was taken, after which 0.2 mL of $KI-I_2$ was added. The samples were mixed and left under refrigeration at 0–4 °C for 16 h. They were then centrifuged at 3100 RCF for 15 min at 0 °C and placed on ice for 1 h. Finally, the supernatant was separated and 9 mL of 1,2-Dichloroethane was added and left at room temperature for 2 h, then 1 mL of the sample was removed using a pipette and the absorbance was read at 365 nm in the spectrophotometer (Thermo Scientific Genesys 10S, Chelmsford, MA, USA). The values obtained were interpolated in a calibration curve using a glycine-betaine standard.

## 2.7. Experimental Design and Data Analysis

All experiments were conducted with a completely randomized design and replicated three times. An analysis with a factorial arrangement was conducted for the development variables recorded for the cultivars × PEG combinations (levels). The data obtained were tested with an analysis of variance (ANOVA) followed by Tukey's test ($p \leq 0.05$), performed using SPSS statistical software (Windows version 22).

## 3. Results

The osmotic potential ($\Psi$s) of the culture medium increased as PEG concentrations increased, obtaining values of $\Psi$s = $-0.16$, $-0.45$, $-0.63$ and $-0.80$ MPa for 0, 3, 6 and 9% PEG, respectively. On the other hand, when evaluating the effect of different PEG concentrations on the in vitro development of sugarcane (*Saccharum* spp. Hybrids) cv Mex 69-290, Mex 79-431, CP 72-2086 and MOTZMex 92-207, significant differences were observed between the different combinations of cultivars and PEG concentrations with regard to the development variables evaluated. Also, we noted an interaction in the factorial analysis in the variables shoot length, number of roots per explant and root length, although this was not observed for the number of shoots per explant, number of leaves

per explant and dry matter. In addition, significant differences were observed for all the development variables (Table 1).

**Table 1.** Effects of different polyethylene glycol (PEG) 6000 concentration on in vitro development of sugarcane (*Saccharum* spp. Hybrids).

| Cultivar | PEG (%) | Number of Shoots per Explant | Shoot Length (cm) | Number of Leaves per Explant | Number of Roots per Explant | Root Length (cm) | Dry Matter (%) |
|---|---|---|---|---|---|---|---|
| Mex 69-290 | 0 | 8.16 ± 0.70 [a] | 8.02 ± 0.32 [a] | 3.24 ± 0.17 [a] | 9.50 ± 0.61 [a] | 3.10 ± 0.15 [a] | 7.57 ± 0.38 [d] |
| | 3 | 7.33 ± 0.42 [a] | 8.17 ± 0.15 [a] | 3.33 ± 0.16 [a] | 9.33 ± 0.61 [a] | 3.16 ± 0.18 [a] | 10.27 ± 1.55 [bcd] |
| | 6 | 4.50 ± 0.22 [bcd] | 7.00 ± 0.28 [abc] | 2.61 ± 0.14 [bcd] | 2.66 ± 0.33 [bc] | 1.04 ± 0.15 [d] | 15.00 ± 0.67 [a] |
| | 9 | 3.66 ± 0.33 [cde] | 5.68 ± 0.23 [e] | 2.10 ± 0.16 [cd] | 1.83 ± 0.30 [c] | 1.12 ± 0.13 [d] | 15.10 ± 0.53 [a] |
| Mex 79-431 | 0 | 7.16 ± 0.30 [a] | 7.36 ± 0.18 [abc] | 2.94 ± 0.18 [ab] | 8.00 ± 0.36 [a] | 3.20 ± 0.16 [a] | 7.29 ± 0.51 [d] |
| | 3 | 4.66 ± 0.33 [bcd] | 6.76 ± 0.19 [bcd] | 2.86 ± 0.19 [abc] | 2.83 ± 0.30 [bc] | 2.20 ± 0.17 [b] | 8.70 ± 0.59 [cd] |
| | 6 | 3.50 ± 0.34 [cde] | 6.56 ± 0.20 [bcde] | 2.26 ± 0.11 [bcd] | 2.00 ± 0.36 [c] | 1.00 ± 0.16 [d] | 14.13 ± 0.25 [ab] |
| | 9 | 2.83 ± 0.30 [de] | 6.38 ± 0.33 [cde] | 2.15 ± 0.22 [cd] | 1.50 ± 0.22 [c] | 1.13 ± 0.17 [d] | 15.01 ± 0.37 [a] |
| CP 72-2086 | 0 | 7.00 ± 0.36 [a] | 7.36 ± 0.15 [abc] | 3.26 ± 0.15 [a] | 7.83 ± 0.40 [a] | 3.03 ± 0.14 [a] | 7.51 ± 0.35 [d] |
| | 3 | 4.80 ± 0.37 [bc] | 6.54 ± 0.24 [bcd] | 2.83 ± 0.20 [abc] | 4.20 ± 0.37 [b] | 2.04 ± 0.20 [bc] | 8.87 ± 0.23 [cd] |
| | 6 | 2.60 ± 0.24 [e] | 6.45 ± 0.63 [cde] | 2.72 ± 0.14 [bcd] | 2.40 ± 0.50 [bc] | 1.28 ± 0.14 [cd] | 14.18 ± 0.96 [ab] |
| | 9 | 2.60 ± 0.24 [e] | 5.95 ± 0.40 [e] | 2.16 ± 0.24 [cd] | 1.80 ± 0.37 [c] | 1.05 ± 0.13 [d] | 15.48 ± 1.17 [a] |
| MOTZMex 92-207 | 0 | 7.00 ± 0.36 [a] | 8.23 ± 0.30 [a] | 3.03 ± 0.16 [ab] | 8.00 ± 0.36 [a] | 3.34 ± 0.10 [a] | 7.82 ± 0.64 [d] |
| | 3 | 4.83 ± 0.30 [bc] | 7.65 ± 0.20 [abc] | 3.07 ± 0.13 [ab] | 4.50 ± 0.42 [b] | 2.17 ± 0.15 [b] | 12.53 ± 0.43 [bc] |
| | 6 | 3.20 ± 0.37 [cde] | 7.51 ± 0.44 [abc] | 2.27 ± 0.23 [bcd] | 3.20 ± 0.48 [bc] | 1.31 ± 0.12 [cd] | 13.56 ± 1.30 [ab] |
| | 9 | 2.20 ± 0.37 [e] | 6.42 ± 0.64 [cde] | 1.90 ± 0.27 [d] | 2.00 ± 0.44 [c] | 1.00 ± 0.13 [d] | 15.78 ± 1.16 [a] |
| *p*-value | | | | | | | |
| *p* (cultivar) | | 0.000 | 0.001 | 0.301 | 0.000 | 0.064 | 0.046 |
| *p* (PEG) | | 0.000 | 0.000 | 0.000 | 0.000 | 0.000 | 0.000 |
| *p* (cultivar × PEG) | | 0.666 | 0.000 | 0.400 | 0.000 | 0.000 | 0.121 |

Values represent the mean ± SE. Means with different letters within a column were significantly different (Tukey, $p \leq 0.05$).

### 3.1. PEG-Induced Osmotic Stress

In general, a gradual reduction in the number of shoots per explant was observed as the PEG concentration increased in cv Mex 79-431, CP 72-2086 and MOTZMex 92-207. The highest number of shoots per explant was found in 0 and 3% PEG treatments of cv Mex 69-290, obtaining 7–8 shoots per explant, while the lowest number of shoots was found in the 6 and 9% PEG treatments of CP 72-2086, with 2.60 and 2.60 shoots, respectively, and in the 9% PEG treatment of cv MOTZMex 92-207, with 2.20 shoots per explant. For the variable shoot length, the longest shoots were obtained in 0 and 3% PEG treatments of cv Mex 69-290, with 8.02 and 8.17 cm lengths, respectively, and 0% PEG in cv MOTZMex 92-207, with an 8.23 cm length, whereas the shortest shoot length was observed with 9% PEG in cv Mex 69-290. and MOTZMex 92-207, with 5.68 and 5.95 cm lengths, respectively. For the variable number of leaves per shoot, the treatments with the highest number of leaves were 0 and 3% PEG in cv Mex 69-290, with 3.34 and 3.33 shoots per explant, respectively, and 0% PEG in cv CP 72-2086, with 3.26 leaves per shoot. For the number of roots, the treatments with the highest number of roots were 0 and 3% PEG in cv Mex-69-290 and 0% PEG in cv Mex 79-431, CP 72-2086 and MOTZMex 92-207, obtaining 7–9 roots per explant, while the lowest number of roots was observed with 9% PEG in cv Mex-69-290, 6 and 9% PEG in cv Mex 79-431, 9% PEG in cv CP-72-2086 and 9% PEG in cv MOTZMex 92-207, obtaining less than two roots per explant. For root length, the treatments with the longest roots were recorded with 0 and 3% PEG in cv Mex-69-290 and with 0% PEG in cv Mex 79-431, CP 72-2086 and MOTZMex 92-207, obtaining 3 cm long roots on average, while the shortest root length was observed in the rest of the PEG treatments in all cultivars evaluated, with roots of 1–2 cm in length. Finally, for the percentage of dry matter, the treatments with the highest dry matter content were 6 and 9% PEG in cv Mex-69-290 and 9% PEG in cv Mex 79-431, CP 72-2086 and MOTZMex 92-207, obtaining on average 15% dry matter, while the

lowest dry matter content was observed in 0% PEG in all evaluated cultivars, with 7% dry matter on average (Figure 1).

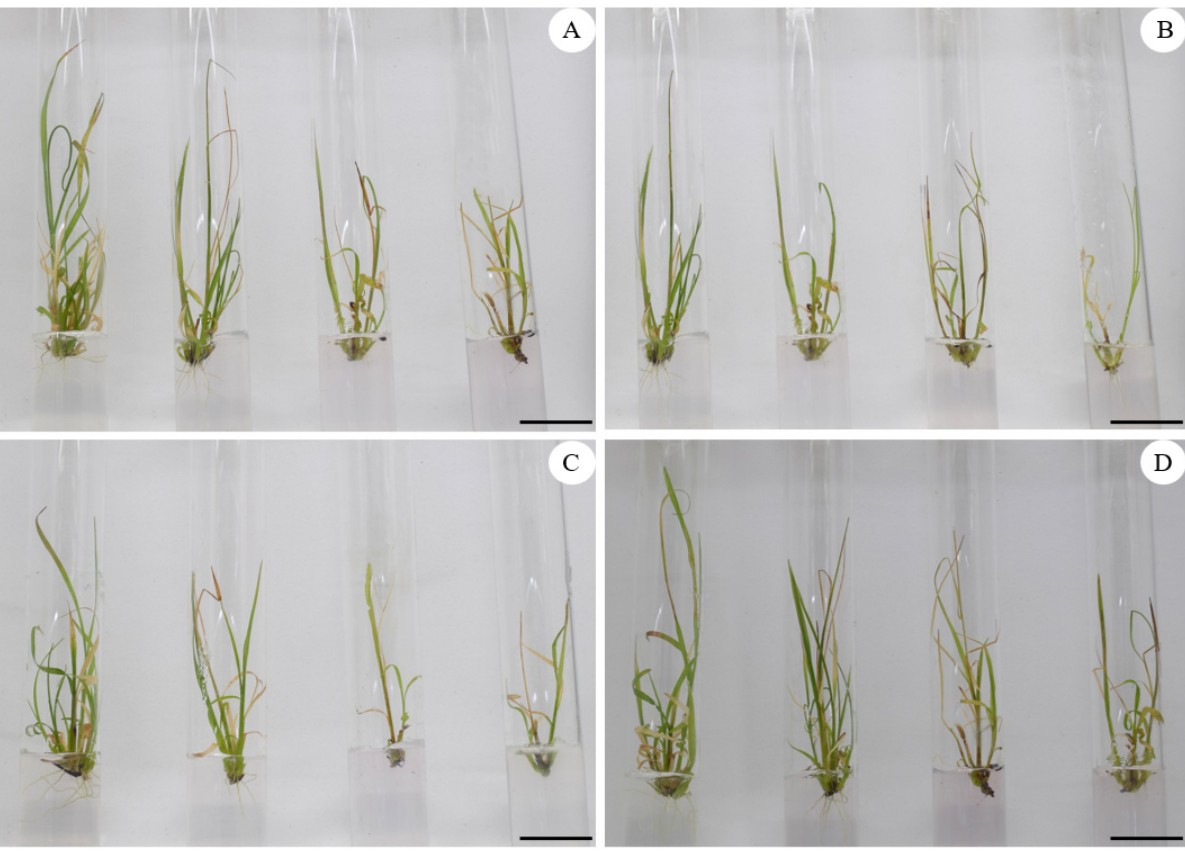

**Figure 1.** Effect of different concentrations of PEG-6000 (0, 3, 6, and 9%, left to right) on sugarcane cultivars (*Saccharum* spp. Hybrids) during in vitro selection. (**A**) Mex 69-290, (**B**) Mex 79-431, (**C**) CP 72-2086 and (**D**) MOTZMex 92-207. Black bar = 2 cm.

### 3.2. Total Protein Content

In general, TP contents decreased with increasing PEG concentration in all evaluated cultivars. In addition, the factorial analysis showed interaction between cultivars and PEG concentrations ($p \leq 0.05$). The highest protein contents were found in 0% PEG treatments without PEG in the evaluated cultivars, obtaining between 12–13 $g^{-1}$ in fresh weight. The lowest protein contents were observed in the highest PEG concentration evaluated (9%) in cv Mex 69-290, with 0.77 mg $g^{-1}$ in fresh weight (Figure 2).

### 3.3. Proline Content

Proline concentration underwent a gradual increase with increasing PEG concentration in all cultivars evaluated. In addition, the factorial analysis showed interaction between cultivars and PEG concentrations ($p \leq 0.05$). The highest Pr content was found in the 9% PEG treatments of the cultivars evaluated, obtaining between 7–8 µmol $g^{-1}$ in fresh weight. The lowest Pr content was found in 0% PEG treatments in all the cultivars evaluated, obtaining between 4.60–4.95 µmol $g^{-1}$ in fresh weight (Figure 3).

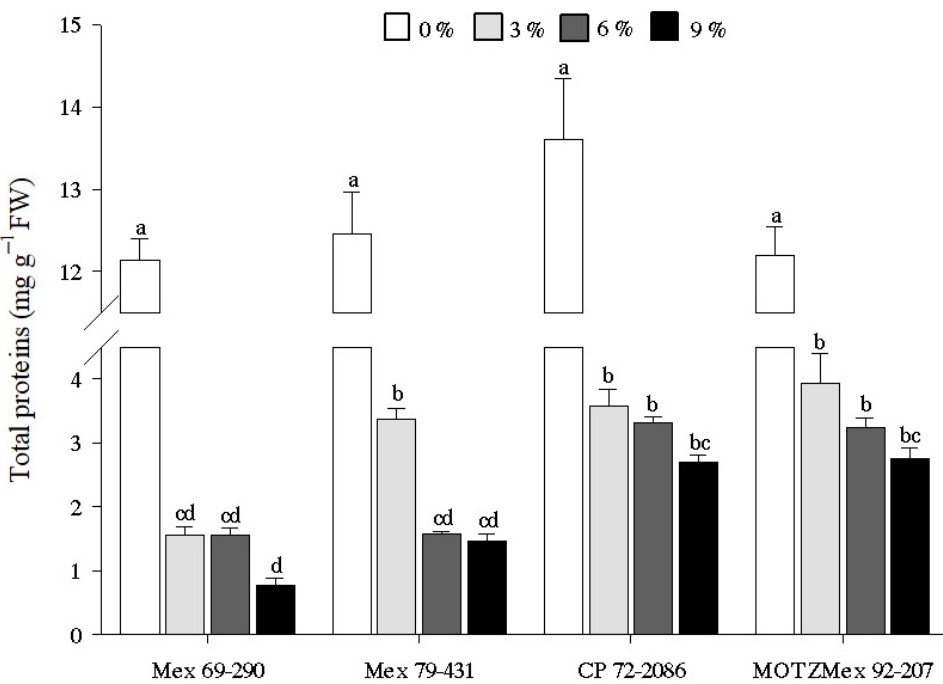

**Figure 2.** Effect of different concentrations of PEG-6000 on total protein content in sugarcane cultivars (*Saccharum* spp. Hybrids). Bars above the columns represent the standard error. Different letters above the columns of each subfigure indicate significant statistical differences among treatments (Tukey, $p \leq 0.05$).

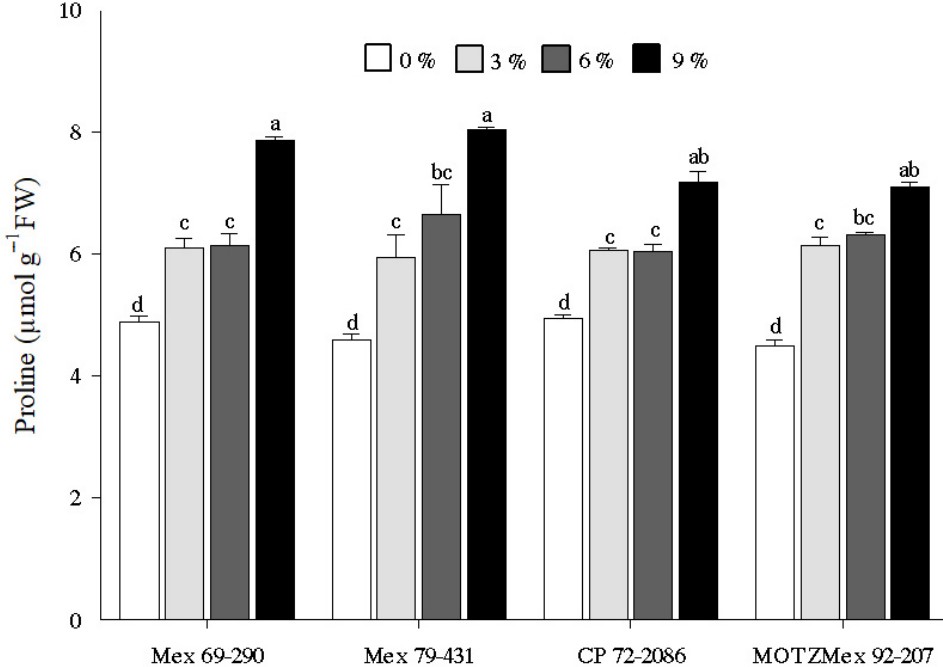

**Figure 3.** Effect of different concentrations of PEG-6000 on proline content in sugarcane cultivars (*Saccharum* spp. Hybrids). Bars above the columns represent the standard error. Different letters above the columns of each subfigure indicate significant statistical differences among treatments (Tukey, $p \leq 0.05$).

### 3.4. Glycine-Betaine Content

The GB content increased with increasing PEG concentrations in all cultivars evaluated. In addition, the factorial analysis showed interaction between cultivars and PEG

concentrations ($p \leq 0.05$). The highest GB content was observed in the treatments with 9% PEG in all cultivars, obtaining between 100–101 µmol g$^{-1}$ dry weight. The lowest GB contents were found in 0% PEG treatments, obtaining between 9.25–11.37 µmol g$^{-1}$ in dry weight (Figure 4).

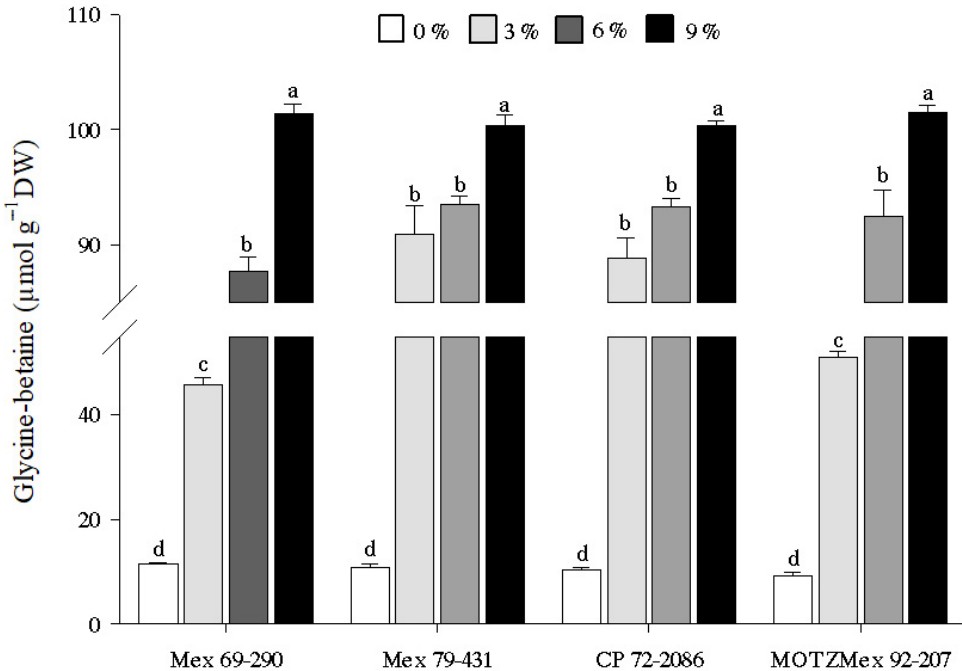

**Figure 4.** Effect of different concentrations of PEG-6000 on glycine-betaine content in sugarcane cultivars (*Saccharum* spp. Hybrids). Bars above the columns represent the standard error. Different letters above the columns of each subfigure indicate significant statistical differences among treatments (Tukey, $p \leq 0.05$).

## 4. Discussion

### 4.1. Effect of PEG on In Vitro Development of Sugarcane Seedlings

The results demonstrate that in vitro development of sugarcane (*Saccharum* spp. Hybrids) is affected under osmotic stress conditions induced by different PEG concentrations. Different PEG concentrations had an effect on the osmotic potential of the culture medium ($\Psi$s = −0.16, −0.45, −0.63 and −0.80 MPa for 0, 3, 6 and 9% PEG). A decrease in the developmental variables evaluated with increasing the PEG percentage was observed in cv Mex 79-431, CP 72-2086 and MOTZMex 92-207; however, cv Mex 69-290 showed no differences in development at 3% PEG. According to a previous study [9], shoot and root length can be used as selection criteria for early detection of water stress-tolerant genotypes. On the other hand, the authors of another study [20] noted that an increase in the medium's water potential significantly affects the in vitro development of sugarcane seedlings. In taro (*Colocasia esculenta* L. Schott) the number of leaves, number of shoots, shoot length and rooting were decreased with 11.8% PEG 6000 [25]. According to Megha et al. [26], the reduction in plant growth is a physiological response to water stress to decrease its metabolism. Shorter shoots increase the development of roots in order to search for water [9]. Similarly, the authors of a previous study [27] stated that the length of the roots is important in plant survival during drought due to their ability to absorb water from the subsoil. A further study [28] suggested that early root elongation is an indicator of drought tolerance.

In our study, cv Mex 69-290 showed no significant differences in development between 0% and 3% PEG treatments, demonstrating tolerance to an osmotic potential of −0.45 MPa. This fact suggests that cv Mex 69-290 could maintain some kind of mechanism to tolerate water stress under in vitro conditions. The increase in the percentage of dry matter in the cultivars evaluated with PEG could be explained by the accumulation of ions, an increase

in the content of compatible osmolytes or the synthesis of proteins and amino acids. Plants accumulate inorganic and organic solutes as the degree of stress gradually increases [29]. Similar results were reported in tomato [30], where using different percentages of PEG 6000 (0%, 20%, 27% and 29.5%) increased dry matter content in all PEG treatments. On the other hand, the authors of [31] reported that, in pea (*Lathyrus sativus*), dry matter increased with increasing osmotic stress. Another study [27] found that, in water stress-induced tomato plants, most of the tolerant genotypes accumulated more matter under osmotic stress with PEG 6000 (4, 8, 12 and 16%). According to the authors of [32], the increase in dry matter during water stress can also be an indicator of drought tolerance.

### 4.2. Quantification of Total Protein Content

Variation in protein content is a tolerance response to water stress [33]. Proteins related to a water stress tolerance response include late embryogenesis (LEA) [34], dehydrins [35], aquaporins [36] and antioxidant enzymes [37], among others. Plants can synthesize these types of proteins to neutralize oxidative stress and osmotic stress [38–40]. Some in vitro studies have reported protein accumulation under PEG-induced water stress in olive (*Olea europaea* L.) [41], wheat [42] and strawberry (*Fragaria × ananassa* Duch.) [43].

In this study, protein content decreased with increasing PEG concentrations. Reduced protein content under in vitro water stress has also been reported in sugarcane callus tissue using PEG 6000 [44]. Similar results were reported in stevia (*Stevia rebaudiana*) [45], palm (*Phoenix dactylifera* L.) [46], ajowan (*Carum copticum* L.) [47] and orchids (*Dendrobium officinale*) [48].

Our results suggest that under the in vitro conditions evaluated, the low protein accumulation in PEG treatments could be explained by two hypotheses: (1) reduced biosynthesis and/or (2) increased protein degradation caused by water stress. Reduced TP content could be associated with the synthesis of specific proteins under water stress that are not significantly quantifiable and low synthesis of those proteins that do not have a specific role in coping with water stress. According to previous studies ([49] and [50]), prolonged osmotic stress can lead to degradation of proteins into amino acids. Amino acids serve as precursors for the biosynthesis of metabolites required for signaling, defense and other functions [51]. Another study [52] found that protein degradation could be the result of increased protease activity and damage to structural or cellular proteins. According to the authors of [53], the alternation of protein synthesis or degradation is one of the fundamental metabolic processes for homeostasis during drought stress.

### 4.3. Proline Content Quantification

Pr accumulation has been attributed to an increase in its biosynthesis and a decrease in its degradation under salt stress and drought conditions [54]. Accumulation of Pr content in plants is a physiological response to water stress [55]. In this regard, the authors of [56] note that Pr is a biochemical indicator used in the selection of water stress-tolerant genotypes. Among the functions of Pr is that of maintaining the stability of structural proteins [57,58], compatible osmolytes [59] and antioxidant activity [60].

In this study, all sugarcane cultivars showed an increase in Pr content when exposed to PEG. This effect confirms that Pr is a biochemical indicator contributing to water stress tolerance mechanisms. Pr accumulation under PEG-induced water stress has been reported in stevia [61,62], rice [63], potato (*Solanum tuberosum*) [64] and mango [13]. In sugarcane, similar results have been reported in callus culture with a semi-solid medium of cv Co86032 in 20% PEG 8000 [65] and when exposing cv Co-86032 callus cultures on a semi-solid medium to 20% PEG 6000 [44]. In this regard, it has been reported [63] that Pr synthesis and accumulation increase in cells subjected to water stress. Pr accumulation may be due to two metabolic pathways: the first involves the synthesis of Pr from glutamic acid [66], while the other pathway is through the deamination of ornithine [57]. These metabolic pathways could occur under water stress conditions.

### 4.4. Quantification of Glycine-Betaine Content

In this study, GB content showed a significant increase when shoots were exposed to PEG in all cv evaluated. To date, GB accumulation in *Saccharum* spp. using PEG under in vitro conditions has not been reported. However, other species accumulate GB under salt, cold and water stress [67,68]. A previous study reported [69], for *Sesuvium portulacastrum* L. calli, a greater increase in GB content with 20% PEG 8000. For date palm, a greater increase in GB concentration in treatments with 15 and 20% PEG 6000 has been reported [70]. Another study [71] on oilseed rape (*Brassica napus*) found an increase in GB content when water stress was induced with different concentrations of PEG 6000. In *Saccharum* spp. Hybrids, GB accumulation has been reported under in vitro salt stress induced by NaCl [72]. Among the functions of GB under water stress conditions are that it acts as a compatible osmolyte [73] and promotes antioxidant activity [74,75]. In addition, it can protect the enzymatic activity of Rubisco and the photosystem II complex during photosynthesis [76,77]. Lack of betaine in cells could cause, among other effects, damage to the cell membrane, proteins and DNA structure, as well as inhibition of photosynthesis [78,79]. GB synthesis under stress conditions is increased by the oxidation pathway of two enzymes, from choline through the unstable intermediate betaine aldehyde [80].

### 4.5. General Discussion

Water stress is caused by scarce water availability, low temperatures and high soil salinity. These conditions reduce $H_2O$ in the cytoplasm of cells, affecting their homeostasis [81]. Water stress generates osmotic and/or oxidative stress, impacting crop development and productivity [8,39]. Various climate change scenarios suggest breeding alternatives for the selection of drought-tolerant sugarcane cultivars. In Mexico, the average sugarcane yield is 74.50 t/ha [82]. This yield, together with alterations in rainfall cycles caused by global climate change, affects the productivity of this crop.

The developmental variables evaluated in vitro made it possible to determine the degree of stress in plants under induced water stress conditions. On the other hand, biochemical determinations of TP, Pr and GB contents acted as biochemical markers of water stress. According to [83], osmotic adjustment mediated by Pr and GB accumulation can be used as early selection markers. However, some exceptions were also described in barley genotypes [84], where higher Pr and GB accumulation were found to be symptoms of stress susceptibility rather than tolerance. In this study, the osmotic potentials were $-0.16$, $-0.45$, $-0.63$ and $-0.80$ MPa, causing a decrease in the relative water content in the culture medium. Of the cultivars Mex 69-290, Mex 79-431, CP 72-2086 and MOTZMex 92-207l, cv Mex 69-290 showed tolerance to $-0.45$ MPa. This result suggests that this cv can be used for rainfed or upland cultivation, whereas cultivars Mex 79-431, CP 72-2086 and MOTZMex 92-2071 could be used in irrigation systems or lowland areas. Through this crop management, together with all required cultivation tasks, these cultivars could reach their productive potential.

## 5. Conclusions

The in vitro water stress-tolerant sugarcane (*Saccharum* spp. Hybrids) screening technique using PEG as a stressing agent is an alternative for the early selection of drought-tolerant cultivars. In our study, cv Mex 69-290 showed tolerance to an osmotic potential of $-0.45$ MPa in 3% PEG 6000. In addition, developmental variables and biochemical determinations can be used as markers for the degree of water stress tolerance. Further research is recommended to study these cultivars in the field.

**Author Contributions:** C.A.H.-P. and J.J.B.-B. designed the experiments, analyzed the data, conducted data interpretation and drafted the manuscript, C.A.H.-P. conducted all the experimental work, F.C.G.-M., J.L.S.-C. and J.J.B.-B. contributed to the conceptualization of the experiment and revising the manuscript. All authors have read and agreed to the published version of the manuscript.

**Funding:** This work was funded by CONACYT, with the project PN 2017-6083 "Innovaciones biotec­nológicas para aumentar la productividad del cultivo de caña de azúcar para el campo mexicano". The funders had no role in the analysis and interpretation of data, or in the writing of the report.

**Institutional Review Board Statement:** Not applicable.

**Informed Consent Statement:** Not applicable.

**Conflicts of Interest:** The authors declare no conflict of interest and the funders had no role in the design of the study; in the collection, analyses, or interpretation of data; in the writing of the manuscript, or in the decision to publish the results.

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
