# Peer review of "In Vitro Screening of Sugarcane Cultivars (Saccharum spp. Hybrids) for Tolerance to Polyethylene Glycol-Induced Water Stress"

_agronomy, doi:10.3390/agronomy11030598_

Round 1
Reviewer 1 Report
This manuscript describes an in vitro selection system for testing water stress tolerance in Saccharum spp. There are different published papers about topic; therefore the novelty of the paper should be highlighted, if it is possible. I miss in the introduction (line 48) more description about works of other authors with sugarcane (cultivars, concentrations of PEG, etc.).
However, the manuscript has a main drawback that need to be deal with. In my opinion data were not properly statistically analyzed. The treatments consist in several PEG concentrations applied to different cultivars in a factorial design. The first thing authors should show (probably in a table) or, at least, indicate within the text is the significance of the different main effects (cultivars and PEG concentrations) as well as the possible interaction. Since the experiments have a structure design, values obtained from different concentrations within cultivar should not be compared by post-hoc tests such as Tukey’s. The total error for each main effect should be partitioned in the errors corresponding to possible orthogonal polynomials contrasts.
In the case of PEG concentrations it should be contrasted the possibility that the data fit a lineal, a quadratic or cubic equation since there are three doses of PEG and the control. Additionally, best fitting equations could be calculated since treatments are equally spaced and therefore orthogonal coefficients can be easily founded. Post-hoc tests, such as Tukey, are frequently abused and misused. It would be too long to explain here why but there are several publications that may be of help to authors in making the right decision on how to analyze their data. For more information they could consult the following references or a statistician.
Compton,M.E., 1994. Statistical methods suitable for the analysis of plant tissue
culture data. Plant Cell, Tiss. Org. Cult. 37, 217-242.
Mize,C.W., Chun,Y.W., 1988. Analysing treatment means in plant tissue culture
research. Plant Cell, Tiss. Org. Cult. 13, 201-217.
Petersen,R.G., 1977. Use and misuse of multiple comparison procedures. Agron.
- 69, 205-208.
Data presented in table 1 would be, therefore, much better presented in figures where the trend of different parameters observed with the doses of PEG could be easily followed. In this figures the theoretical fitting equations could be also presented if they are finally calculated.
I miss the results of Osmotic potential of the culture medium with different PEG concentration. This results just are indicated in the general discussion.
In my opinion the conclusions are not supported by the results because all cultivars showed biochemical determinations too higher in all PEG concentrations compared to the control (0%). Only some developmental variables did not change in the cultivar cv. Mex 69-290 when the lowest PEG concentration was applied.
Minor concerns:
Lines 188 and 189: Deleted
Reviewer 2 Report
Questions for authors:
Were the sugarcane cultivars used in the experiments selected randomly or on the basis of some specific criteria? Is something known about water stress tolerance/sensitivity of these cultivars (from previous studies or from agricultural practice)?
Based on which were the tested PEG concentrations/osmotic potential levels chosen?
Specific comments on the text:
- Introduction
The correct name of family is „Poaceae“ not „Poacea“.
- Materials and methods
2.1 Plant material and in vitro establishment
Instead of „indolacetic acid“ should be stated correctly „indoleacetic acid“ or more precisely „indole-3-acetic acid“.
The methods used for the detection of proteins, proline and glycine betaine are classical, but in my opinion precise enough to assess the diferent responses of various genotypes to osmotic stress.
- Discussion
4.1 Effect of PEG on in vitro development of sugarcane seedlings
The claim that the increase in plant dry matter under the influence of osmotic stress can be explained, among other things, by protein synthesis is in conflict with the results of the analysis of the total protein content in plants (of course, the synthesis of specific water stress related proteins is increased but the overall synthesis of proteins decreases).
4.3 Proline (Pr) content quantification
Yes, accumulation of proline is generally considered to be higher in osmotic stress tolerant plant species but some exceptions were also described where higher proline accumulation was found to be rather symptom of stress susceptibility than tolerance (e.g. Lutts et al., 1996; Chen et al., 2007 etc.). This should be also discussed.
Reviewer 3 Report
The paper „In vitro Screening of Sugarcane Cultivars (Saccharum spp. Hybrids) for Tolerance to Polyethylene Glycol-Induced Water Stress“ is focused on the application of PEG-induced water stress screening technique under in vitro conditions for testing of sugarcane hybrid cultivars for drought stress tolerance. Four cultivars - Mex 69-290, CP 72-2086, Mex 79-431 and MOTZMex 92-207 were investigated in this respect by treatment with different concentrations of PEG 6000 (0, 3, 6 and 9% w/v) using Murashige and Skoog semi-solid culture medium.
In other studies on sugar cane, the stress tolerance was investigated by molecular profiling using RAPD technique, yield attributes and biochemical parameters or using mutation induction and NACl selection of embryogenic calli. By contrast to above mentioned studies, the novelty of this presented study consists in evaluation of different developmental variables, such as shoot length, number of leaves, root length, number of roots, percentage of dry matter (DM), total protein (TP), proline (Pr) and glycine-betaine (GB) contents under in vitro selection system using PEG 6000 as a selective agent.
The results obtained in this study are useful for determination of the degree of stress in plants under induced water stress conditions and biochemical determinations of TP, Pr and GB contents can act as biochemical markers of water stress as early selection markers. These results can be used as a foundation for effective crop management which enable to sugarcane cultivars to reach their productive potential.
The paper is well organized and summarizes the obtained results very clearly and logically. The paper can be published after minor revision according to the following comments:
Line 48: Give more details on the results of these studies in sugarcane and reason why you decided to apply this technique for testing of your hybrid cultivars.
Line 62: For in vitro culture establishment, isolated meristems were used or just apical buds? Please, give more details on initial explants. If isolated meristems were used, describe their preparation (isolation) and size.
Line 76: What was duration of the subculture?
Round 2
Reviewer 1 Report
Firstly, the novelty of the paper is not highlighted. Although more description about works of other authors with sugarcane have been included in the introduction, the originality of the present work is not remarkable. Authors should explain why they choose the studied cultivars, their interest, etc.
Authors have not changed the statistical analysis. They only include the significance of the different main effects (cultivars x PEG concentrations) and interaction. In my opinion it is not enough, since the analysis is not correct. Values obtained from different concentrations within cultivar are still compared by Tukey’s post-hoc test. Dunnett’s test is better in this case because values should be compared with control. Please, consult a statistician.
The effect of different PEG 6000 concentration on in vitro development is still presented in Table 1. In my opinion much better presented in figures where the trend of different parameters observed with the concentration of PEG could be easily followed.
